# Best Practices for Comprehensive Annotation of Neuropeptides of *Gryllus bimaculatus*

**DOI:** 10.3390/insects14020121

**Published:** 2023-01-25

**Authors:** Takako Mochizuki, Mika Sakamoto, Yasuhiro Tanizawa, Hitomi Seike, Zhen Zhu, Yi Jun Zhou, Keisuke Fukumura, Shinji Nagata, Yasukazu Nakamura

**Affiliations:** 1National Institute of Genetics, Research Organization of Information and Systems, 1111 Yata, Mishima, Shizuoka 411-8540, Japan; 2Department of Integrated Biosciences, Graduate School of Frontier Sciences, The University of Tokyo, 5-1-5 Kashiwanoha, Kashiwa, Chiba 277-8562, Japan

**Keywords:** *Gryllus bimaculatus*, neuropeptides, genome annotation, draft genome annotation, functional annotation

## Abstract

**Simple Summary:**

We identified the neuropeptides and their genomic loci on the draft genome sequences of *Gryllus bimaculatus*. These annotations were additionally assigned to the draft genome annotation. This addition to the draft genome annotation improved the convenience of research by consolidating the knowledge of neuropeptides, such as the sequence information and functional annotation, which could not be obtained without searching each article, into the draft genome annotation. This contributed to the infrastructure for facilitating genome-wide research using high-throughput sequencing technology.

**Abstract:**

Genome annotation is critically important data that can support research. Draft genome annotations cover representative genes; however, they often do not include genes that are expressed only in limited tissues and stages, or genes with low expression levels. Neuropeptides are responsible for regulation of various physiological and biological processes. A recent study disclosed the genome draft of the two-spotted cricket *Gryllus bimaculatus*, which was utilized to understand the intriguing physiology and biology of crickets. Thus far, only two of the nine reported neuropeptides in *G. bimaculatus* were annotated in the draft genome. Even though de novo assembly using transcriptomic analyses can comprehensively identify neuropeptides, this method does not follow those annotations on the genome locus. In this study, we performed the annotations based on the reference mapping, de novo transcriptome assembly, and manual curation. Consequently, we identified 41 neuropeptides out of 43 neuropeptides, which were reported in the insects. Further, 32 of the identified neuropeptides on the genomic loci in *G. bimaculatus* were annotated. The present annotation methods can be applicable for the neuropeptide annotation of other insects. Furthermore, the methods will help to generate useful infrastructures for studies relevant to neuropeptides.

## 1. Introduction

Genome annotation is vital infrastructural data that can support various studies for the organism under study, as well as for comparative understanding using other organisms. Draft genome annotations are designed and utilized to acquire an overview of the genes on the genome. Although most of the representative genes are well-addressed in the draft annotation, some groups of genes that are important for extensively differentiated studies are often missing or only annotated imperfectly. These imperfect annotations resulted from the transcripts located in specific tissues, expressed at limited levels and stages, or consisting of very short sequences. These may cause difficulties in conducting comprehensive genome annotation.

Neuropeptides exert regulatory mechanisms on various physiological and biological processes in insects, including circadian rhythms, courtship, development, feeding, olfaction, and reproduction [1]. Structurally, most neuropeptides are small molecules composed of fewer than 30 amino acids [2]. These neuropeptides are first translated from mRNA into precursor peptides and proteins, and then proteolytically cleaved and modified into mature peptides [3,4]. To function in their physiological roles, mature peptides are secreted into the extracellular environment and bind to the appropriate receptors, which are primarily the G protein-coupled receptors (GPCRs) featured with seven transmembrane alpha-helices [3].

Comparative studies with other insect neuropeptide sequences were performed by searching the transcriptome sequences assembled from RNA-seq [5,6,7,8] to identify neuropeptides in the two-spotted cricket, *Gryllus bimaculatus*. Although analysis through de novo transcriptome assembly reveals the high coverage of transcripts, information on genomic loci was not available. The information on genomic loci can expand the variety and convenience of gene analysis; most expression analysis tools for RNA-seq and single-cell RNA-seq (scRNA-seq) require the genomic loci, gene regions, as well as their adjacent sequences inspected in the transcription factor analysis, and gene copy number estimation is not feasible without the information related to genomic loci.

In 2021, the draft genome sequence of *G. bimaculatus* and its genome annotation was released [9]. A total of nine neuropeptides were previously reported in crickets, namely: adipokinetic hormone (AKH) [7], AKH/corazonin-related peptide (ACP) [7], allatostatin A (Ast A) [10], allatostatin B (Ast B) [11,12], corazonin [6], elevenin [6], myosuppressin [13], pigment dispersing factor (PDF) [14], and sulfakinin [15]. However, only two (Ast A and sulfakinin) were annotated in the genome assembly.

In this study, we identified neuropeptides and their genomic loci on the draft genome of *G. bimaculatus* in combination with the homology-based search and manual curation by experts on insect neuropeptides. Of the 43 neuropeptides, 41 neuropeptides and 32 neuropeptide genomic loci were identified in this analysis. These procedures complemented the incomplete annotation of the neuropeptides of *G. bimaculatus*, and they will be the basis for various studies in the future. In particular, the method described in the present study is applicable for comparative studies on other insect neuropeptide annotations.

## 2. Materials and Methods

### 2.1. Overview of Identification Method for Neuropeptides

The procedure for neuropeptide identification is shown in Figure 1.

In neuropeptide identification, we first assessed the reference neuropeptide sequences derived from the previous reports of transcriptomic and peptidomic investigations of other insects [16,17,18,19,20,21,22,23]. Using these reference neuropeptide sequences, we provisionally detected neuropeptide sequences of *G. bimaculatus* using our in-house database derived from *G. bimaculatus* RNA-seq datasets. This database was constructed in our previous study [12]. The resulting sequences were then explored by performing homology searches against the transcriptome sequence sets from the draft genome annotation, the reference mapping-based annotation, and the de novo transcriptome assembly-based annotation. The candidate neuropeptides were selected via curation after multiple alignments with the reference neuropeptide sequences. Neuropeptide annotations were allocated to the GFF file of the draft genome annotation. These new annotated neuropeptide sequences were validated with public RNA-seq data (Figure 2).

This figure shows an overview of the method used to identify neuropeptides and their loci in this study. Textboxes with numbering represent the indicated processes. 1. Neuropeptide sequences were collected from previous studies of *G. bimaculatus* and other insects. 2. The neuropeptide sequences were used as seed sequences for the homology search against transcriptome datasets, the draft genome annotation, reference mapping-based (StringTie-based) annotation, and de novo transcriptome assembly based (Trinity-based) annotation. 3. The candidate neuropeptide sequences, which were detected via homology searching, were manually curated with multiple alignments. 4. These neuropeptide sequences were validated with the public RNA-seq data. 5. These annotations were added to the GFF file of the draft genome annotation.

### 2.2. Identification of Seed Neuropeptide Sequences

The cDNA and translated protein of each neuropeptide were selected by searching for the sequences of closely related orthopteran species against the in-house database of *G. bimaculatus* transcripts constructed in previous reports [12]. Among the 43 neuropeptides that were reported in insects, 35 neuropeptides were provisionally determined as the reference neuropeptide sequences, which are the mature, precursor, or cDNA sequences (Appendix A). In particular, the sequences of ACP [7], Ast B [12], corazonin [6], elevenin [6], myosuppressin [13], and PDF [14] were used as seed sequences as described in previously published studies. These selected proteins, except for sequences published in previous studies, were submitted with BLASTP [24] on the NCBI website (https://blast.ncbi.nlm.nih.gov/Blast.cgi (accessed on 16 July 2022)) with the clustered nr database to confirm homology with the sequences of closely related orthopteran species. These precursor proteins and their amino acid sequences deduced from cDNA were confirmed to belong to that they belong to appropriate protein clusters (Appendix A).

### 2.3. RNA Sequencing Data

*G. bimaculatus* RNA-seq datasets were obtained as follows: brain—subesophegeal ganglion—thoracic ganglia, fat body, and corpora cardiaca samples were collected, and total RNA was extracted with Trizol (Invitrogen Life Technologies/ThermoFisher Scientific, Waltham, MA, USA) and phenol-chloroform. Library construction and sequencing was provided as a custom service of Eurofins Genomics K. K. (Tokyo, Japan). The polyA captured libraries were subjected to paired-end 2 × 100 bp sequencing on the HiSeq 2500 platform (Illumina, San Diego, CA, USA). Apart from these experiments, total RNA was extracted from the anterior midgut and ovary. Library construction and sequencing was provided as a custom service of Macrogen Japan Corp. (Kyoto, Japan). The polyA captured libraries were subjected to paired-end 2 × 100 bp sequencing on the NovaSeq 6000 platform (Illumina, San Diego, CA, USA). These datasets were deposited to the Sequence Read Archive (DRR358356-DRR358364).

### 2.4. Construction of Transcript Datasets

Raw reads are trimmed using fastp (v0.20.0) (-cut_front, -cut_tail option) [25]. Trimmed reads were all merged and processed subsequently. Transcript sequences were assembled with the trimmed reads by two methods: genome-guided and de novo transcriptome assembly. In the genome-guided transcriptome assembly, the trimmed reads were mapped to the *G. bimaculatus* draft genome sequence using HISAT2 (v2.2.1) (-rna-strandness RF, -dta option) [26]. They were assembled into transcripts using StringTie (v2.1.4) [27]. Then, TransDecorder (v5.5.0) [28] was used to extract protein coding regions from StringTie resulting data with the alignment files from hmmscan (http://hmmer.org/, v3.3.1 (accessed on 25 July 2020)) against Pfam [29] and BLASTP (v2.10.1) (-evalue 1 × 10^−5^ option) against Uniref90. In the de novo transcriptome assembly, trimmed reads were assembled into transcripts using Trinity (v2.11) [30]. Next, the transcript sequences and deduced amino acid sequences were obtained using Trinotate (https://trinotate.github.io/ (accessed on 9 April 2020)) (v3.2.1) (LOAD_swissprot_blastp, LOAD_swissprot_blastx, and LOAD_pfam option) for extraction of protein coding regions and functional annotations. The SAM/BAM files were manipulated by SAMtools (v1.11) [31]. The sequences of the gene regions defined in the GFF format file were obtained using GffRead (v0.12.1) [32].

### 2.5. Search for Neuropeptide-Related Loci

The three gene annotation sets, i.e., the draft genome annotation, StringTie-based annotation, and Trinity-based annotation, were used to identify neuropeptide loci. In the case where at least one of the mature peptide, precursor, and cDNA sequences were available as seeds, they were aligned through the three gene annotation sets by BLASTP or BLASTN. The gene models were selected as candidate neuropeptide sequences if they matched one of the following criteria: at least one of the mature peptides was perfectly matched; the alignment length of the precursor sequence was at least 70%; and the alignment length of cDNA sequence was at least 60%. In the case of the neuropeptides for which precursor sequences of *G. bimaculatus* were not identified, the precursor sequences of other species were used as the seed. The gene models were selectively utilized as candidate neuropeptide sequences if the alignment length of the precursor sequence was at least 50%. When the corresponding annotation was not found by the abovementioned method, the reference cDNA was aligned against draft genome sequences using BLASTN to estimate the genomic loci, and gene models in the loci were subjected to curation. The genome sequence was constructed by long reads and reflected more accurate repeat regions and segmental duplications. Therefore, when the same sequence hit multiple loci, these were included in the manual curation. In the search for RYamide, in addition to the homology search of cDNA against draft genome sequences, the amino acid sequence with the mature peptide sequence motif “GSRYGKR” [33] was used to determine its candidates. Finally, the candidates were curated over the alignments among the neuropeptides by MAFFT (v7.490) (-clustalout, -reorder option) [34]. Amino acid and nucleotide sequences were both used for these alignments. Manual curation was performed as follows: the position of each mature peptide within each alignment was displayed to confirm the conserved amino acid residues and the processing position. If necessary, the open reading frame (ORF) of the transcript was arranged using seqkit (v2.2.0) [35], i.e., when no mature peptide was found despite matching the cDNA sequence, the reading frame of the cDNA was changed to search for the mature peptide. Visualization of the gene models on the draft genome sequences were performed using JBrowse (v1.16.1) [36].

### 2.6. Validation of Neuropeptide Genomic Loci

The curated neuropeptides were validated by expression analysis using the publicly available RNA-seq data of *G. bimaculatus* (SRR14026720-SRR14026726) [37]. Raw reads were initially trimmed using fastp (v0.20.0) (-cut_front, -cut_tail option). The abundances of neuropeptides were quantified using kallisto (v0.48.0) [38] with the trimmed reads and transcriptome sequences, including the draft genome annotation and newly annotated neuropeptides. The latter process was performed using R (v4.0.5). Briefly, transcripts per kilobase million (TPM) of the neuropeptide transcripts were normalized with a z-score using the genefilter package (v1.72.1) [39]. The heatmap was depicted using the pheatmap package (v1.0.12) [40].

### 2.7. Preparation for Data Release of New Neuropeptide Annotation

To determine the loci of neuropeptides from Trinity-based annotation, the CDS sequences were mapped to the draft genome using GMAP (v2021.03.08) [41]. The genomic sequences of mapped regions were translated to amino acid sequences. The genomic loci were determined if the amino acid sequences were identical to the amino acid sequence generated by Trinotate. The gene models for which genomic loci could not be determined were added to the GFF file of the draft genome as new contigs. Transcript and amino acid sequences were added to the FASTA file of the draft genome. The neuropeptides from StringTie-based annotation were also added to the GFF and fasta files of the draft genome. New loci were numbered from ‘GBI_30000’.

## 3. Results

### 3.1. Neuropeptides Identification

The reference neuropeptide sequences were qualified as seeds to find their genomic loci and used for the subsequent steps. Only 8 of 43 neuropeptides were not determined as any of the mature, precursor, or cDNA sequences. These eight neuropeptides were searched using the precursor sequences of other insects as seeds. The candidate neuropeptides were detected from the draft genome annotation, StringTie-based annotation, and Trinity-based annotation with these mature, precursor, and cDNA sequences (Appendix A). The candidate neuropeptides were manually curated on multiple alignment among the precursor sequences of the neuropeptides and candidates (see Appendix A).

As a result, among the 43 neuropeptides, 41 were identified from the draft genome annotation, StringTie-based annotation or Trinity-based annotation. Although 32 of 41 neuropeptide-coding loci were identified on the draft genome, we were unable to find the remaining 9 neuropeptide-coding loci on the draft genome, and they were placed on the Trinity-assembled contigs (Table 1). Additionally, 18 of 32 genome-annotated neuropeptides were identified from the gene models via the draft genome annotation, and 15 of the 18 neuropeptides had the functional annotation of each neuropeptide in the draft genome annotation.

Neuroparsin precursor (NPP) was examined with two different seed sequences, NPP1 and NPP2, which partially share homologous sequences. Despite the two distinctive NPP seeds, manual curation revealed that NPP1 and NPP2 were observed as one locus merged together. Unexpectedly, the functional domains of Neuroparsin were detected in two loci: one of the StringTie-based sequences that was identical to the seed sequence and the other in the draft genome annotation that was similar but not identical (84% in alignment length) to the seed. The former was designated as NPP isoform 1, whereas the latter was designated as NPP isoform 2. AKH, myosupressin, proctolin, and PDF were not identified by homology searching with reference neuropeptide sequences (mature, precursor, and cDNA). Therefore, gene models and loci were identified by BLASTN searches of cDNA sequences against genomic sequences. To obtain their provisional genomic loci, Trinity-based annotation was mapped to the draft genome using GMAP, and overlapping gene models in the loci where cDNA sequences were aligned by BLASTN were investigated. Subsequently, the gene models in Trinity-based annotation were picked out from the same locus. Their amino acid sequences were determined from the genomic sequences via manual curation. The amino acid sequence of myosupressin [13] was unavailable from the Trinity-based annotation. Hence, the ORF of the Trinity contig was changed with “seqkit translate”, and the gene model was determined. For trissin, the gene models of the draft genome annotation (GBI_01877-RA, GBI_01877-RB) were adopted because of the more highly conserved sequence of the mature peptides than the gene models that were detected by the homology search with the precursor sequence of the closely related orthopteran species (GBI_01745-RA, GBI_01745-RB). Although natalisin was well conserved in cDNA sequence (STRG.7468.1.p4), the amino acid sequence deduced from the gene model did not have a sequence of the mature peptide and did not have high similarity to the reference amino acid sequences. This discrepancy might be attributable to mutations (e.g., indels and SNPs) in the draft genome sequence. Therefore, the mutations in the draft genome sequence were loss of function mutations. Several identified sequences with multiple subtypes in the transcripts were observed. These transcripts might be derived from alternative splicing variants. Indeed, ion transport peptide and ion transport peptide-like (ITP and ITPL) are known variants that are derived from alternative splicing [45]. Similarly, the current analyses provided the transcript variants possibly derived from ITP and ITPL.

To determine the locus of neuropeptides from Trinity-based annotation, such as ACP, Ast A, allatostatin CC (Ast CC), allatotropin, CCHamide-1, CCHamide-2, corticotropin releasing factor-like diuretic hormone (CRF/DH), kinin (Leucokinin), myosuppressin, neuropeptide F1a (NPF1a), and neuropeptide F1b (NPF1b), their CDS sequences were mapped to the draft genome, but the genomic loci were not determined.

### 3.2. Validation of Curated Neuropeptides

The neuropeptides were validated by the expression analysis with the public RNA-seq data of the whole body of *G. bimaculatus* (SRP311541) (Appendix A) [37]. The expression levels of annotated neuropeptide precursor genes were depicted by the heatmap accounting for TPMs of transcripts (Figure 2). For each of the gene models for trissin and elevenin, trissin (GBI_01877-RB) and elevenin (STRG.16982.2.p3), the transcript per million (TPM) values were 0 in all samples, which were exceptional cases in the heatmap. Cluster analyses of the transcriptional levels of each neuropeptides revealed three substantially different transcriptional patterns: higher levels at an early stage, steady increasing level according to post embryonic development, and a steep increase possibly owing to molting and eclosion. The putative splicing variants of neuropeptides were observed to have a differential expression pattern, indicating that the assigned transcriptional subtypes of neuropeptides were precisely annotated in the current study and that all subtypes had the possibilities of differentiated transcriptional levels and localization.

## 4. Discussion

In total, 41 of the previously reported 43 neuropeptides in crickets and other insects were identified. EFLamide and Tachykinin were not identified in the genomic sequences or in the Trinity-assembled contigs in this study.

Currently, comprehensive neuropeptides and bioactive peptides were demonstrated in several insect species. Comparisons of the lists of neuropeptides revealed that several missing neuropeptides can be observed differently according to insect species. For example, allatotropin and ACP are absent in *D. melanogaster*, whereas corazonin is absent in *Tribolium castaneum*. Such neuropeptide loss does not occur in accordance with the phylogeny, but with occasional events during the evolutionary diversification of each species [43,46,47]. Similarly, the current study did not show the presence of a tachykinin-like peptide in *G. bimaculatus*, although its homologous receptor genes were observed in the genomic and transcriptomic data. Such orphan ligands or receptors can be observed in other invertebrates [46].

In case of tachykinin-like peptide receptors, several distinctive ligands exhibit affinity in *D. melanogaster* and *Bombyx mori*. Indeed, natalisin binds to the tachykinin receptor in *D. melanogaster*; ITP also binds to the tachykinin receptor in *B. mori* [48]. According to these cases, it can be implied that there are naturally occurring coincidences in which several structurally independent neuropeptides can share some receptors. Meanwhile, the current study did not exhibit the presence of EFLamide. Recent studies reported a fragment sequence encoding EFLamide in the closely related orthopteran species, *Locusta migratoria*, which belongs to the same order as *G. bimaculatus* [47,49]. Similarly, a fragment sequence (KHQRNFLKGIRSISQIVYSARIVRNLGEFLGK), with an EFLamide-like peptide, was present in the genomic data of *G. bimaculatus*; however, the transcriptomic data did not provide any contigs including this partial sequence.

The 41 neuropeptides identified in the current study were validated using the expression analysis with the publicly available data. The transcript levels of each neuropeptide were substantially clustered in three transcriptional patterns, possibly reflecting the growth stage. Although trissin and elevenin had more than one splicing variant, one of each (trissin: GBI_01877-RB; elevenin: STRG.16982.2.p3) was not expressed among all samples. The alternatively spliced variants in neuropeptide transcripts were frequently observed so far. The transcriptional levels of the variants often exhibit specific spatiotemporal expression patterns [50]. The apparent absence of alternatively spliced transcripts encoding trissin and elevenin might be due to the relatively low expression level within the data sets of the analyzed periods throughout cricket growth and development. In contrast, several neuropeptides were observed as a single subtype, although other insect species have various subtypes. For example, alternatively spliced variants of allatotopin are observed in lepidopteran species [51,52]. Similarly, alternative splicing variants in diuretic hormone 31 (CT/DH) are also observed. However, the current study illuminated only single subtypes of allatotropin and CT/DH.

In the present study, we also measured the transcriptional levels of all annotated neuropeptide precursors using publicly available datasets (Figure 2). At present, there are few examples in which the transcriptional levels of neuropeptides were comprehensively measured and reported. This might be because general neuropeptides are spatiotemporally expressed in individually specific patterns. For example, the transcriptional pattern of CT/DH precursors are variously located according to the splicing variants, possibly causing differently altered levels of these transcripts throughout growth and development [50]. Similarly, most reports dealt with specific tissues, sometimes very small tissues, at a specific developmental stage, probably because most neuropeptides are believed to be expressed in a spatiotemporally limited pattern. As the present study demonstrated, RNA derived from the whole body can provide transcriptional patterns of all annotated neuropeptides, and it would be helpful to address the developmental traces of neuropeptide transcriptional levels.

Amino acid and cDNA sequences encoding both mature and the precursor of neuropeptides and bioactive peptides in other insect species are available from open resources, such as NeuroPep and DINeR [2,53]. BLASTP with mature peptides might not be applicable for the deep search of other insect species. Most results obtained by BLASTP led to the transcripts with other functions, owing to the shortage in the length of mature peptides even though they had highly conserved sequences (alignment% ≥ 90%). To avoid this difficulty, we proposed to search first with precursor sequences and then with the mature sequence.

Consequently, several patterns were found in which the neuropeptide precursor genes were not annotated correctly in the draft genome annotation. For example, the genomic locus in Ast B assigned in the draft genome annotation was correct, but the gene model does not show its existence. In the course of this detailed annotation, we found a new splicing variant at the genomic locus of Ast B that may code the precursor peptide. In another case, although CRF/DH had the correct genomic locus, a mutation in the genome sequence prevented proper amino acid translation; thus, the gene function was annotated as an unknown protein in the previously reported annotation. In contrast, the strategy used in this study covered amino acid sequences that were translated in the ORF prediction stage of the Trinity-assembled cDNA, but not all the translations provided the correct translation reading frame (as the setting to adopt the longest ORF was adopted). In this case, there was a discrepancy between the relatively higher homology in the cDNA sequences and the lower or little homology of the deduced amino acid sequences. Therefore, finding the processes of these loci or transcripts for neuropeptide precursors, including AKH, PDF, and proctolin, for example, was complicated in terms of their discrepancies. One of the solutions to this problem was to change the reading frame of the cDNA. In such cases, it was helpful to return to the homology of the nucleotide sequences of transcripts and the genomic sequences by BLASTN and to re-confirm the nucleotide sequences and the corresponding amino acid sequences with JBrowse (Figure 3).

A case in which the open reading frame (ORF) prediction from a Trinity-assembled contig failed to predict the original amino acid translation. (A) In the left part, the yellow rectangle shows the genomic position of the reference cDNA of proctolin aligned to the draft genome sequence by BLASTN. The narrow black rectangle shows the location where the exon encoding the mature peptide of proctolin is found. An enlarged view is shown on the right. The black arrow in the enlarged figure indicates the direction of the deduced amino acid sequence, and the black rectangle indicates the deduced amino acid sequence. (B) The amino acid sequence of the proctolin precursor; the highlighted string indicates the mature peptide of proctolin. (C) The contig aligned by GMAP to the position shown in (B) and translated using seqkit command with a different reading frame. Each sequence is the longest ORF starting with the first methionine in that reading frame. The ORF derived from the reading frame −3 is the longest among all calculated ORFs. The mature peptide of proctolin was not found in the longest ORF of this contig. In the frame-altered translation, the amino acid sequence of the proctolin precursor was matched completely, and their mature peptide also appeared.

The goal of this work was to assign additional annotations for neuropeptide precursors to the *G. bimaculatus* genome sequence to facilitate gene expression analysis by high-throughput sequencing technology. This objective was achieved by homology searches using known homologous/related sequences and by expert confirmation and manual curation of the presence or absence of mature sequences. In several cases, mature sequences or cleavage sites were missing, even when the sequence homology computed by BLAST was high, reminding us of the limitation of our strategy solely based on sequence homology. Meticulous curation by experts would fill in the gaps between homology-based annotations and true annotations; indeed, we identified almost all the neuropeptide precursor loci on the *G. bimaculatus* genome sequence. However, this is only an achievement of annotation on the current *G. bimaculatus* genome sequence. With newer convenient technologies, such as HiFi reads by Pacific Biosciences, more accurate genome sequences will be published in the future and the annotation will need to be revised. Nonetheless, this work is significant because it re-annotated and validated neuropeptides from knowledge only available within a small and limited community where it is sometimes inaccessible from outside its scientific field. The genome annotation of non-model organisms is often unsophisticated. However, we were able to present an example of how researchers can enhance the annotation quality by refining the annotations of their own research area and redistributing them, even if the quality of the draft genome annotation is insufficient. If researchers with varied expertise re-annotate their own area, it is possible to conduct research using enriched annotation information. In the future, creating an environment in which researchers can share information while refining existing annotations will be necessary. Furthermore, the methods described in this study can be applied to the neuropeptide annotation of other insects if neuropeptide sequences of the target organism or other species are available. In particular, our strategy, which primarily relies on transcriptome sequences, is applicable even to organisms with relatively large genome sizes, such as orthopteran insects.

The current draft genome annotation includes several receptors; however, it does not cover all of them. We are ready to annotate the receptors for neuropeptides, as well as the neuropeptide annotations, as performed in this study. Hopefully, further studies will clarify the correspondence with the neuropeptides and enrich the annotations.

## 5. Conclusions

We identified the genomic loci of neuropeptides on the draft genome of *G. bimaculatus*. These annotations were added to the draft genome annotation. This approach can be applied to other insects. Furthermore, we present a valid case study that updates the draft genome annotation with the findings of individual gene studies to enrich the research infrastructure in non-model organisms.

## Figures and Tables

**Figure 1 insects-14-00121-f001:**
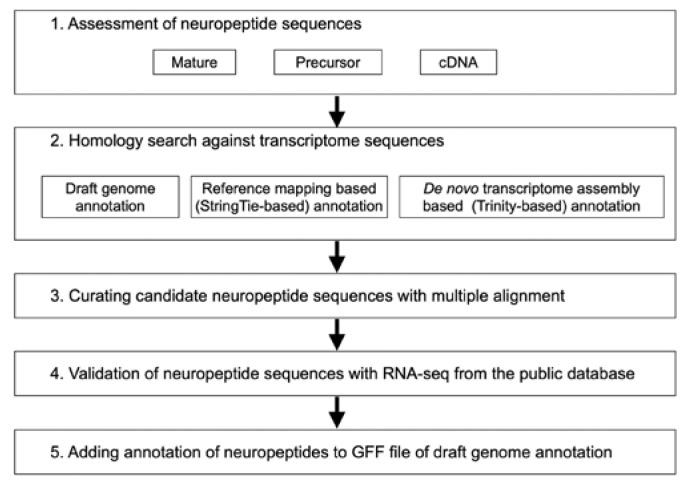
Overview of identification method for neuropeptides.

**Figure 2 insects-14-00121-f002:**
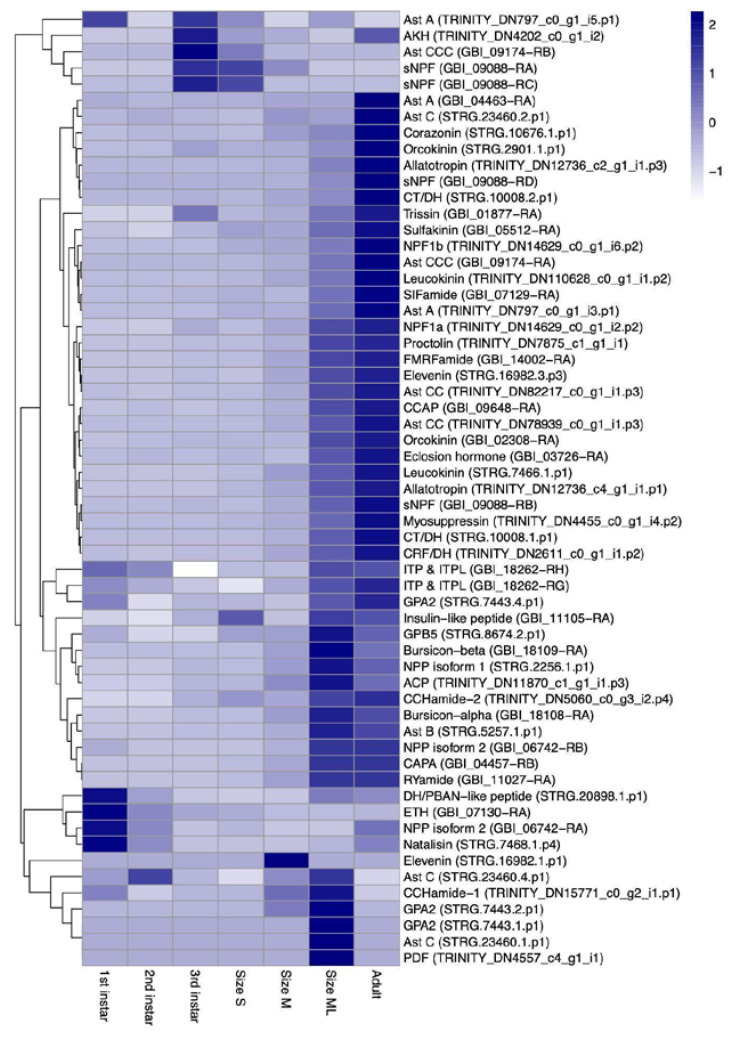
Heatmap of the normalized TPM value for neuropeptides. The expression analysis was performed with the public RNA-seq data for the whole body of *G. bimaculatus* (SRP311541) to validate the neuropeptide annotations. Trissin (GBI_01877-RB) and Elevenin (STRG.16982.2.p3) were removed from the heatmap because they had transcript per million (TPM) values of 0 in all samples. The sizes S, M, and ML indicate the size of wingless juvenile; 6–10 mm (size S), 10–15 mm (size M), and 15–20 mm (size ML).

**Figure 3 insects-14-00121-f003:**
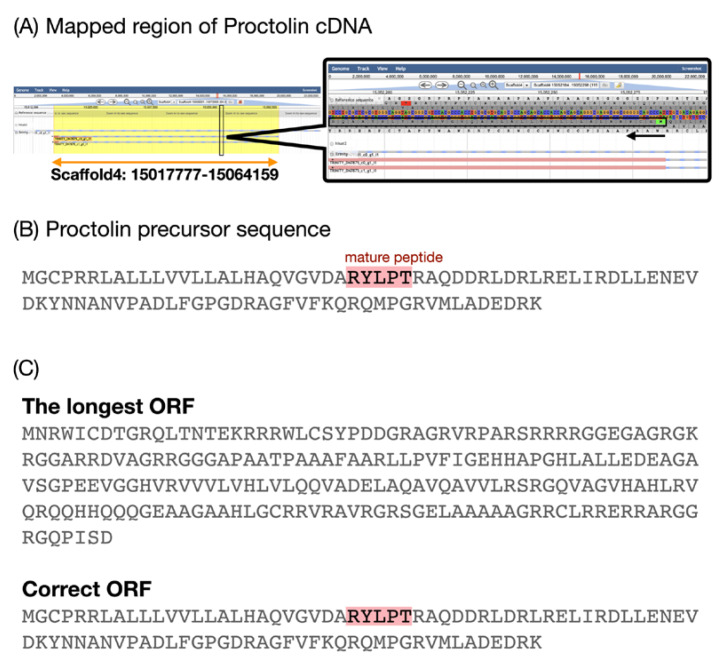
Procedure for finding the genomic locus of the proctolin precursor. (**A**) Mapped region of proctolin cDNA. (**B**) The amino acid sequence of the proctolin precursor. (**C**) The translated amino acid sequences with different reading frames from the mapped region.

**Table 1 insects-14-00121-t001:** *Gryllus bimaculatus* neuropeptides.

Neuropeptides	ID	Locus	*D. melanogaster*	*B. mori*	Method ^(1)^
Adipokinetic hormone (AKH)	GBI_30800-RA (TRINITY_DN4202_c0_g1_i2)	Scaffold206: 697999…707834 (+)	+	+	T
AKH/Corazonin-related peptide (ACP)	GBI_32000-RA (TRINITY_DN11870_c1_g1_i1.p3)	−	−	+	T
Allatostatin A (Ast A)	GBI_32100-RA (TRINITY_DN797_c0_g1_i3.p1), GBI_32200-RA (TRINITY_DN797_c0_g1_i5.p1) GBI_04463-RA	GBI_04463-RA ^(2)^ Scaffold20: 11377336…11377923 (−)	+	+	D,T
Allatostatin B (Ast B)	GBI_04462-RB (STRG.5257.1.p1)	Scaffold20: 10912781…10975738 (−)	+	+	S
Allatostatin C (Ast C)	GBI_31000-RA (STRG.23460.1.p1), GBI_31000-RB (STRG.23460.2.p1), GBI_31000-RC (STRG.23460.4.p1)	Scaffold437: 197807…241684 (−)	+	+	S
Allatostatin CC (Ast CC)	GBI_32300-RA (TRINITY_DN78939_c0_g1_i1.p3), GBI_32400-RA (TRINITY_DN82217_c0_g1_i1.p3)	−	+	+	T
Allatostatin CCC (Ast CCC)	GBI_09174-RA, GBI_09174-RB	Scaffold60: 395418…593331 (+)	−	−	D
Allatotropin	GBI_32500-RA (TRINITY_DN12736_c4_g1_i1.p1), GBI_32600-RA (TRINITY_DN12736_c2_g1_i1.p3)	−	−	+	T
Bursicon-alpha	GBI_18108-RA	Scaffold254: 577937…579923 (−)	+	+	D
Bursicon-beta	GBI_18109-RA	Scaffold254: 587165…588284 (+)	+	+	D
CAPA	GBI_04457-RB	Scaffold20: 9936553…9977407 (+)	+	+	D
Corazonin	GBI_30500-RA (STRG.10676.1.p1)	Scaffold62: 3436728…3448223 (−)	+	+	S
Crustacean cardioactive peptide (CCAP)	GBI_09648-RA	Scaffold65: 3361094…3402145 (+)	+	+	D
CCHamide-1	GBI_32700-RA (TRINITY_DN15771_c0_g2_i1.p1)	−	+	+	T
CCHamide-2	GBI_32800-RA (TRINITY_DN5060_c0_g3_i2.p4)	−	+	+	T
Diapause hormone (DH/PBAN-like peptide)	GBI_30900-RA (STRG.20898.1.p1)	Scaffold262: 1299901…1301218 (+)	+	+	S
Diuretic hormone 31 (CT/DH)	GBI_08745-RB (STRG.10008.1.p1), GBI_08745-RC (STRG.10008.2.p1)	Scaffold56: 1634530…1842684 (+)	+	+	S
Corticotropin releasing factor-like diuretic hormone (CRF/DH)	GBI_32900-RA (TRINITY_DN2611_c0_g1_i1.p2)	−	+	+	T
Ecdysis triggering hormone (ETH)	GBI_07130-RA	Scaffold39: 4483375…4511070 (+)	+	+	D
Eclosion hormone	GBI_03726-RA	Scaffold15: 14005275…14019193 (+)	+	+	D
Elevenin	GBI_30700-RA (STRG.16982.1.p1), GBI_30700-RB (STRG.16982.2.p3), GBI_30700-RC (STRG.16982.3.p3)	Scaffold153: 2142142…2267228 (+)	−	+	S
EFLamide	−		−	−	−
FMRFamide	GBI_14002-RA	Scaffold134: 2784690…2789881 (−)	+	+	D
Glycoprotein hormone A2 (GPA2)	GBI_30100-RA (STRG.7443.1.p1), GBI_30100-RB (STRG.7443.2.p1), GBI_30100-RC (STRG.7443.4.p1)	Scaffold34: 5739999…5757709 (−)	+	+	S
Glycoprotein hormone B5 (GPB5)	GBI_30400-RA (STRG.8674.2.p1)	Scaffold43: 6774211…6779917 (−)	+	+	S
Insulin-like peptide	GBI_11105-RA	Scaffold84: 3089694…3140894 (−)	+	+	D
Ion transport peptide and ion transport peptide-like (ITP and ITPL)	GBI_18262-RG, GBI_18262-RH	Scaffold259: 893509…930810 (−)	+	+	D
Kinin (Leucokinin)	GBI_30200-RA (STRG.7466.1.p1), GBI_33200-RA (TRINITY_DN110628_c0_g1_i1.p2)	GBI_30200-RA) ^(3)^ Scaffold34: 8543640…8544062 (−)	+	+	S,T
Myosuppressin	GBI_33000-RA (TRINITY_DN4455_c0_g1_i4.p2)	−	+	+	T
Natalisin	GBI_30300-RA (STRG.7468.1.p4)	Scaffold34: 8647546…8652008 (−)	+	+	T
Neuropeptide F1a (NPF1a)	GBI_33300-RA (TRINITY_DN14629_c0_g1_i2.p2)	−	+	+	−
Neuropeptide F1b (NPF1b)	GBI_33100-RA (TRINITY_DN14629_c0_g1_i6.p2)	−	+	+	T
Neuroparsin precursor (NPP) isoform 1	GBI_01783-RB (STRG.2256.1.p1)	Scaffold7: 15634784…15638238 (+)	−	+	S
Neuroparsin precursor (NPP) isoform 2	GBI_06742-RA, GBI_06742-RB	Scaffold35: 7759086…7774792 (+)	−	+	D
Orcokinin	GBI_02308-RA, GBI_02308-RB (STRG.2901.1.p1)	Scaffold9: 16015120…16184531 (−)	+	+	D, S
Pigment dispersing factor (PDF)	GBI_30600-RA (TRINITY_DN4557_c4_g1_i1)	Scaffold130: 1613008…1616041 (−)	+	+	T
Proctolin	GBI_30000-RA (TRINITY_DN7875_c1_g1_i1)	Scaffold4: 15016980…15064115 (−)	+	+	T
RYamide	GBI_11027-RA	Scaffold83: 5210571…5235250 (−)	+	+	D
Short neuropeptide F (sNPF)	GBI_09088-RA, GBI_09088-RB, GBI_09088-RC, GBI_09088-RD	Scaffold59: 3780407…4100907 (+)	+	+	D
SIFamide	GBI_07129-RA	Scaffold39: 4382305…4384876 (+)	+	+	D
Sulfakinin	GBI_05512-RA	Scaffold28: 3070275…3081238 (−)	+	+	D
Tachykinin	−	−	+	+	−
Trissin	GBI_01877-RA, GBI_01877-RB	Scaffold8: 2096488…2133885 (−)	+	+	D

The data for *D. melanogaster* and *B. mori* were referenced from [23,42,43,44]. ^(1)^ The method was described as an annotation method. ‘D’ indicates draft genome annotation, ‘S’ indicates StringTie-based annotation, and ‘T’ indicates Trinity-based annotation. ^(2)^ The genomic loci for GBI_32100-RA and GBI_32200-RA were not identified in the draft genome. ^(3)^ The genomic locus for GBI_33200-RA was not identified in the draft genome.

## Data Availability

The data presented in this study: *Gryllus bimaculatus* draft genome sequence and gene annotation with neuropeptides are available in figshare (https://figshare.com/articles/dataset/Supplementary_table/21524097 (accessed on 24 November 2022)). RNA-seq data sets using in this study were obtained from Sequence Read Archive (DRR358356-DRR358364 for gene annotation, SRR14026720-SRR14026726 for validation).

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
