# Peer review of "Best Practices for Comprehensive Annotation of Neuropeptides of Gryllus bimaculatus"

_insects, 2023, doi:10.3390/insects14020121_

Round 1

Reviewer 1 Report

The authors report on the annotation of neuropeptides in the cricket genome. I would like to point out a few things about this paper.

- As the authors point out, neuropeptides have short sequence lengths. Have the authors adjusted the parameters of the tool to map short sequences? If the parameters described are different from the defaults, it would be good to explain the reasons for the changes as your own innovations.

- Orthopteran insects, such as crickets and grasshoppers, generally have large genome sizes. Did the authors need to address the large genome size in any way in this study? Also, do you need to take any action to apply this method to other orthopteran insects with even larger genome sizes?

- Suggestions for discovering novel neuropeptides that may not have been previously known as transcripts, especially when applied to other insects, may improve the attractiveness of this technique.

After all, what do you think was the cause of the neuropeptides being only partially annotated in crickets? The quality of the draft genome? Lack of transcripts? Poor quality of previous annotations? Limitations of existing annotation processes? I think the quality of this paper would be improved if you could clarify the problems and how the authors overcame them.

Reviewer 2 Report

This paper clearly describes how to annotate non-model organisms for which annotation information is lacking. Analysis has been carried out with a particular focus on neuropeptides, and no major problems have been observed in the experimental procedures.

I would like to point out two points that caught my attention when I was peer-reviewing.

Although it is a minor misspelling,

2.7 in l.171 should be revised to 2.6.

Also, what kind of options did Trinotate use for precursors with a short number of amino acid residues, such as neuropeptides? (l.142-143)

If you change the default parameters, please describe them so that the readers can easily reproduce the method of this paper.

Reviewer 3 Report

General comments:

The authors have taken up the Gryllus bimaculatus draft genome assembly to annotate the neuropeptides in this study. The draft genome annotation contains only two of the nine reported neuropeptides in G. bimaculatus. The de novo transcriptome assembly and annotation can identify more neuropeptides; however, this approach cannot locate this info in the genomic loci. So, the authors used de novo transcriptome assembly, reference mapping, and manual curation to annotate 41 out of 43 neuropeptides reported in the insects and located 32 in the genomic loci. The draft genome annotation presents representative genes but doesn't necessarily include genes from specific tissues or categories. Hence, as neuropeptides are significant for various physiological and biological processes, authors argue that this additional annotation will be of great use to many researchers, and the annotation method described can be replicated for other insect genomes.

I appreciate the author's effort to carry out this analysis. The annotation generated could be helpful to other researchers in the field. However, the manuscript needs revision for its clarity and grammar use. At times it was difficult to follow what authors were trying to convey.

The introduction could be more assertive with added information on why it is necessary to annotate these neuropeptides. How could the scientific community benefit from these curated resources? General introduction and importance of mature, precursor, and cDNA sequences of neuropeptides and how these different sequence categories could help for better annotation. The last introduction paragraph could be shortened.

The materials and methods section needs to clarify why specific approaches were used and how they were implemented in detail (more specific comments below). Also, the order of the subsections should be rearranged for clarity (more comments below). Some methodology parts need a more detailed description for clarity and reproducibility (specific comments below). Please move all the described methodologies from the results to this section. The discussion could be improved by detailing how this approach could be replicated in other organisms.

Specific comments:

Line 59-61: Can the authors also state why it is preferable to define genomic loci for various analyses mentioned?

Line 62-68: This could be clearer. Something like:

A total of nine neuropeptides were previously reported in crickets, namely: adipokinetic hormone (AKH) (REF),…….however, only two (Ast A and sulfakinin) were annotated in the genome assembly.

Line 72: reference needed for "most neuropeptides are expressed in brain and fat body"

Line 87-90: Can the authors add the details on the names, types, sequences of the neuropeptides, and the source organism they got from the literature as a query in this study in the supplementary materials?

Line 88: what do the cDNA sequences of neuropeptides refer to, and how are they different from the mature and precursor neuropeptides? Can you please elaborate?

Line 90-92: How did the authors "provisionally detected" neuropeptides? mapped/blast references to RNA-seq reads/assembly or other methods? Please describe.

Line 91-92: Briefly describe the RNA-seq database used in this study. Is it mRNA-seq or total-RNAseq, or other kinds? These RNA-seq reads were used for StringTie-based and Trinity-based annotation, so please give metrics of the RNA-seq data used. How many reads, sequencing type/method, and sample info?

Line 92: reference "13" refers to the RNA-seq database curated for another study. So, please refer to the original source of the RNA-seq database.

 Line 93: "…confirmed by homology search…" what do you mean by confirmed? Confirmed they are present in each database or confirmed the different types? It needs to be clearer.

Line 92- 96: How were the stringTie-based annotation, Trinity-based annotation, and annotated transcripts from the genome assembly used to confirm by homology ssearch? Please elaborate and make it clear.

Line 95-96: Can you briefly elaborate on how was the stringTie based annotation performed? What aligner was used (HISAT/Tophat…), annotation pipeline, and other info as necessary. Same thing with Trinity-based annotation. Did you use the trinotate pipeline or some other pipeline? Trinity does de novo assembly, not annotation. Please make it clear for the readers so they can reproduce your results.

Line 96-98: Can the authors elaborate on what they mean by "curating the presence of mature peptides"? What was the aligner used? Please mention the parameters other than the default.

Line 99: what was the software used to map the sequences? Do you mean "Trinity-based assembly or annotation? (See comments above).

Line 100: Did the authors manually allocate new annotation to the draft genome annotation? If they have used any software or scripts, please mention them.

Line 100-102: How was the validation done? By observing the different expressions of neuropeptides in different developmental stages of the insect?

Line: 104- 112: Figure 1. It can be made better. At least the space and alignment of the text inside the box could be done better, for example. The first box could be smaller in width and the text "Assessment of neuropeptide sequences" could be centrally aligned. Same for the other boxes as well. The Figure caption should be more descriptive.

Line 113-115: how did the authors select closely related insect species? Within family/order/or other groups?

Line 114: Don't need to use acronyms for a single word "database" as DB. Replace it in all occurrences.

Line 115: Refer to the original RNA-seq database paper, not the secondary one. Reference 13 also refers to another paper as the source of the RNA-seq reads.

Line 117: Recheck reference 27. It refers to MMseq2 software, not the database mentioned.

Line118-121: why only six neuropeptides (ACP, Ast B, corazonin, elevenin, myosuppressin, and PDF) were used as seeds and left others mentioned above? Please clarify.

Line 121 – 130: Was the first RNA-seq data from a previous study and the second RNA-seq data generated during this study? Mention clearly, if both are from a single experiment, why were different library prep and sequencing carried out? For the library preparation and sequencing part, which was carried through a service provider, please mention the kit that was used for the library preparation, the ribo-depletion method used, and what kind of sequencing was done (paired-end/single-end, fragment length, and other parameters as relevant).

Maybe the curation of datasets should be mentioned before describing how the datasets were analyzed. Hence, I have comments above for lines 91-92. Similarly, the construction of transcript datasets should be mentioned earlier in the methodology section, which would cover some of my questions above.

Line 132-133: Mention the parameters used for –cut_front and –cut_tail options? Did the authors also use a quality threshold to QC the raw reads?

Line 147-170: How did the authors deal with multi-mapping? Please clarify.

Line: 174-176: What reference sequences were used to quantify the reads? Only the annotated neuropeptide sequences from the draft assembly were extracted to map the reads.

Line 176-177: what do the authors mean by the latter process was performed in R? please clarify.

Line 193-200: Move to the methods section. Also, further down, please remove the method part from the results and state in the materials and method section with all the details needed.

Line 222: Please mention the percentage similarity.

Line 259 – 272: Is a similar expression pattern found in other organisms through the developmental stages? This could be described in the discussion section.

Line 274-278: describe what the size S, M, and ML stands for.

Line 287-288: what do the authors mean by occasional events in each species? References, please?

Line 339-343: might fit better to describe in the method section.
